# Is it Possible to Predict MGMT Promoter Methylation from Brain Tumor MRI Scans using Deep Learning Models?

**Numan Saeed**                                          NUMAN.SAEED@MBZUAI.AC.AE
**Shahad Hardan**                                      SHAHAD.HARDAN@MBZUAI.AC.AE
**Kudaibergen Abutalip**                    KUDAIBERGEN.ABUTALIP@MBZUAI.AC.AE
**Mohammad Yaqub**                              MOHAMMAD.YAQUB@MBZUAI.AC.AE
*Mohamed Bin Zayed University of Artificial Intelligence, Abu Dhabi, UAE*

**Editors:** Under Review for MIDL 2022

## Abstract

Glioblastoma is a common brain malignancy that tends to occur in older adults and is almost always lethal. The effectiveness of chemotherapy, being the standard treatment for most cancer types, can be improved if a particular genetic sequence in the tumor known as MGMT promoter is methylated. However, to identify the state of the MGMT promoter, the conventional approach is to perform a biopsy for genetic analysis, which is time and effort consuming. A couple of recent publications proposed a connection between the MGMT promoter state and the MRI scans of the tumor and hence suggested the use of deep learning models for this purpose. Therefore, in this work, we use one of the most extensive datasets, BraTS 2021, to study the potency of employing deep learning solutions, including 2D and 3D CNN models and vision transformers. After conducting a thorough analysis of the models' performance, we concluded that there seems to be no connection between the MRI scans and the state of the MGMT promoter.

**Keywords:** Radiogenomics, Glioblastoma, MGMT promoter, Deep learning

## 1. Introduction

Glioblastoma multiforme is the most malignant and aggressive brain tumor, constituting 60% of adult brain tumors (Taylor et al., 2019). $O^6$-Methylguanine-DNA methyltransferase (MGMT) is a DNA repair enzyme that reduces the effects of alkylating chemotherapeutic agents on tumor cells, leading to a poor response to temozolomide (Taylor et al., 2019). MGMT promoter methylation, or MGMT gene silencing, is a key prognostic factor in predicting the patient's chemotherapy response. Since temozolomide increases the patient's survival rate, the methylation status could contribute to clinical decision (Weller et al., 2010). Patients whose MGMT promoter is methylated report a median survival rate of 21.7, compared to 12.7 months for patients with unmethylated MGMT. The information concerning molecular and genetic alterations of gliomas is usually obtained via invasive procedures, such as biopsy or open surgical resection. This is time and effort consuming from the care provider level and increases the infection risk from a patient level.

The field of radiomics aims to analyze the relationship between clinical diagnoses and genetic characteristics. Identification of genetic properties of tumors could be more effective in creating a personalized treatment plan for each patient (Riemenschneider et al., 2010). Radiomics derives features from medical images to quantify the brain tumor phenotype.

Thus, novel non-invasive approaches are being proposed in order to predict the methylation status from magnetic resonance images (MRI).

There are several studies that showcased the ability of predicting the MGMT promoter status through deep-learning solutions (Chang et al., 2018; Yogananda et al., 2021). However, based on other studies, MGMT promoter is not a reliable prognostic factor for the temozolomide response (Egaña et al., 2020), and cannot be predicted from MRI scans (Han and Kamdar, 2018; Mikkelsen et al., 2020).

In this study, we investigate the ability of state-of-the-art machine learning models to classify the MGMT promoter methylation status. First, we pre-process the MRI scans and then use them to implement three different approaches: applying 2D and 3D convolutional neural networks (CNN), a self-supervised learning (SSL) technique, and vision transformers (ViT). Then, we follow our implementation results with a discussion on the efficacy of these models to predict the MGMT promoter methylation status.

## 2. Related Work

Many attempts involving deep learning methods have recently reported appealing results in classifying MGMT promoter methylation status. Authors in (Chang et al., 2018) have trained a CNN with residual connections to classify the methylation status using T2, FLAIR, T1-weighted pre-and postcontrast MRI scans and achieved a mean accuracy score of 83% on 5-fold cross-validation. MRIs of either low or high-grade gliomas and their corresponding genetic information were obtained from The Cancer Imaging Archives (TCIA) (Clark et al., 2013) and The Cancer Genome Atlas (TCGA) (Tomczak et al., 2015). In another work by (Yogananda et al., 2021), they introduced an MGMT-net, T2WI-only network based on 3D-Dense-UNets, for determining MGMT methylation status alongside segmenting tumors. They report a mean cross-validation accuracy of 94.73% across 3 folds with sensitivity and specificity scores of 96.31% and 91.66%, respectively. The data were from TCIA and TCGA databases. Similar to (Chang et al., 2018), but by using full T2 images performed at Mayo Clinic with recorded MGMT methylation information, Korfiatis et al. (Korfiatis et al., 2017) used ResNet50 and demonstrated an accuracy of 94.90% on a test set.

Despite the success of conventional CNNs in the discussed context above, recent studies raise some concerns regarding MGMT methylation's predictive value and the feasibility of solving this problem using deep learning techniques. Han et al. (Han and Kamdar, 2018) have used a recurrent CNN model that produced a test accuracy of 62% with precision and recall of 0.67. In the Brain Tumor Radiogenomic Classification challenge (Baid et al., 2021; Menze et al., 2015; Bakas et al., 2017), which was hosted by the Radiological Society of North America (RSNA) and the Medical Image Computing and Computer-Assisted Interventions (MICCAI) conference, participants were not able to achieve more than 0.62 AUC score, even though the most extensive dataset for this task was provided. In a recent clinical study (Egaña et al., 2020), epigenetic silencing of MGMT promoter did not help predict response to TMZ in a cohort of 334 patients diagnosed with glioblastoma or high-grade glioma. The study indicates no relationship between methylation of MGMT promoters and overall survival rates. The authors in (Mikkelsen et al., 2020) investigated from a clinical perspective the associations between this predictive biomarker and several radiological and histopathological features in patients with dehydrogenase (IDH) wild-type glioblastomas.

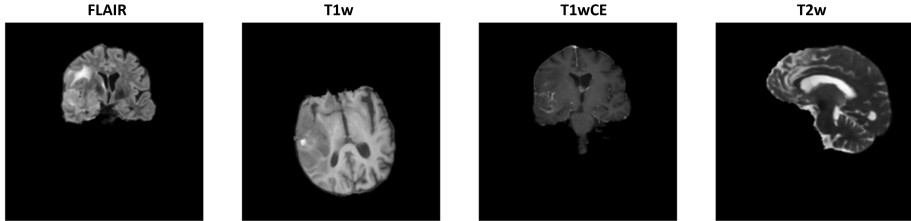

Figure 1: A sample of four random slices from different MRI modalities.

There were no associations between investigated features, including MRI scans characteristics, overall survival, and MGMT status. Authors suggest that the methylation status of MGMT cannot be non-invasively predicted from MRI features.

## 3. Dataset

The dataset used in this study is from the Brain Tumor Radiogenomic Classification challenge (Baid et al., 2021) that includes multi-parametric MRI (mpMRI) scans for 585 glioblastoma patients having 348,641 scans. It was split to 80% training data and 20% testing data. Patients with IDs [00109, 00123, 00709] were removed from the dataset due to issues present in their images. The patients belong to two classes: methylated MGMT and unmethylated MGMT. The dataset is balanced, containing 307 methylated cases and 278 unmethylated cases. Images are in DICOM format that comes with an associated header including clinical information such as modality, orientation, and MRI machine-specific details. Each patient has images in four different modalities: T1-weighted pre-contrast (T1), T1-weighted post-contrast (T1wCE (Gadolinium)), T2-weighted (T2), and T2 Fluid Attenuated Inversion Recovery (T2-FLAIR). The number of slices for each patient differ. MRI images come in three orientations, including coronal, axial, and sagittal. In all the experiments in this manuscript, we use FLAIR or T1wCE type images as the tumor appears bright and easier to distinguish; tumor is thought to encode the information concerning the MGMT promoter state. Figure 1 shows four random slices displaying the four modalities and the existence of different orientations for the same patient. The brain images provided are already preprocessed by being co-registered to the same anatomical template (SRI24), resampled to a uniform isotropic resolution, and skull stripped. The Brain Tumor Radiogenomic Classification challenge dataset includes images from various collections such as the TCIA (Clark et al., 2013) public collection of TCGA-GBM, ACRIN-FMISO-Brain collection (ACRIN 6684)(Gerstner et al., 2016; Kinahan et al., 2018), and other public and private datasets. The following preprocessing steps were performed in this study for all the experiments: 1) Resampling the scans to the axial plane for consistency. 2) For 3D experiments, extracting the same number of slices for each patient. If any further preprocessing steps were performed for an experiment, they would be mentioned in the relevant sections.

## 4. Experiments

### 4.1. Are 2D and 3D CNNs capable of predicting the MGMT promoter status?

CNNs have proven to be effective in various medical imaging applications, including classification. In the first experiment, variations of 2D CNN architecture based on deep residual learning network (ResNet) (He et al., 2015) with 18, 34, 50 layers, and dense connections (DenseNet) (Huang et al., 2018) with 121, 161 layers were used to train the models. Such networks allow to train deeper models without accuracy degradation and address the issue of vanishing gradients. For the 3D CNN model experiment, we implemented 3D Efficient-Net to utilize its ability to capture the information along the depth dimension of the MRI scans. Furthermore, an ensemble network was designed to combine FLAIR and T1wCE modalities: inputs were passed through two branches of the network (ResNet18 without fully connected (FC) layers), and feature maps were either stacked or added before the FC layer. The figure and other details about this experiment can be found in Appendix A. The MRI scans were further preprocessed: 1) cropping the brain part, 2) resizing to three image resolutions for different experiments: $128 \times 128$, $256 \times 256$, $512 \times 512$, 3) 3D contrast limited adaptive histogram equalization (Amorim et al., 2018). In 2D CNN-based classification, additional filtration steps were performed to exclude slices without the tumor.

The last few experiments with CNN models were focused on the region of interest (ROI) - based classification. Segmentation masks for 574 patients were provided in Task 1 of The RSNA-ASNR-MICCAI BraTS 2021 challenge (Baid et al., 2021). Slices were selected based on the segmentation masks to extract only the tumor region, which was resized to image resolutions of $128 \times 128$ and $32 \times 32$. The small image size was used to examine the same approach described in (Chang et al., 2018). The authors used $32 \times 32 \times 4$ input resolution, where four corresponds to stacking different MR modalities, to train custom ResNet comprised of four residual blocks with additional minor modifications. For further details, see Appendix A.

For an extensive evaluation of the networks mentioned above, hyperparameters were tweaked in the following way: the number of epochs varied from 15 to 30, weight decay was set to 0.01 or was not used. For random initializations, the seed was not fixed. Adam optimizer and step-based learning rate scheduling were used. Settings for other hyperparameters and results of the main experiments can be found in Table 1. Overall, selected models either strongly exhibited overfitting behavior or could not capture the relationship between MRI features and MGMT promoter status. It was observed that the training loss fluctuated around the value of 0.7. The rest of the experiments are available in Appendix A. Possible reasons might include noise in the data and insufficient model complexity. More sophisticated network architectures were considered in the following sections. As for the preprocessing steps, we did not notice any positive impact on the performance; hence they were not used for the later experiments to keep the pipeline simple.

### 4.2. Can self-supervision benefit the CNNs in prediction?

Due to the scarcity of labeled images in the medical field (Azizi et al., 2021), SSL methods are used to leverage the large number of unlabeled datasets. In this experiment, we use SimCLR (Chen et al., 2020), a contrastive-based SSL algorithm, to pre-train a ResNet18

Table 1: Results of the main experiments with CNNs. Pret: (Pretrained where 0 and 1 are for Random and ImageNet weights model initialization), Aug: Augmentations. * - as two channels, LR: Learning rate, BS: Batch size.

| Model | Pret. | Modality | Img Res. | Aug. | LR | BS | Train AUC | Val. AUC |
|---|---|---|---|---|---|---|---|---|
| ResNet34 | 0 | FLAIR | 256 | Yes | 1e-2 | 128 | 0.83 | 0.54 |
| ResNet50 | 1 | FLAIR | 128 | Yes | 1e-2 | 128 | 0.61 | 0.58 |
| DenseNet121 | 1 | FLAIR | 128 | Yes | 1e-2 | 128 | 0.63 | 0.58 |
| 3D-EfficientNet | 0 | FLAIR | 256 | No | 1e-3 | 4 | - | 0.64 |
| Custom ResNet | 1 | Stacking | 128 | Yes | 1e-3 | 128 | 0.76 | 0.63 |
| ROI Custom ResNet | 0 | Combined* | 32 | No | 1e-2 | 16 | 0.70 | 0.53 |

model with a proxy task and then finetune it on the downstream task dataset. BraTS 2020 dataset (Menze et al., 2015) was used for the proxy task in the pretraining stage. The BraTS 2020 dataset includes MRI scans of four modalities belonging to 369 patients. The modalities T1, T1w, T2w, and FLAIR, were all included in the proxy task data to maximize the model's understanding of the variations in brain MRI. We extracted 80,720 slices and resized them to $128 \times 128$, disregarding those containing an inconsiderable view of the brain. The pretraining task used a learning rate of 0.1, batch size of 256, and the SGD optimizer with a momentum of 0.9. The augmentations applied are random vertical and horizontal flips, random rotations, and random Gaussian smoothing.

We took 20 slices from each patient for the downstream task and used them as a 20 channel 2D input to the model. The 20 images are of the type FLAIR, in the axial orientation, and resized to $128 \times 128$. They were chosen by taking the slice containing the maximum cutaway of the brain as the central slice. The augmentations implemented on the downstream task are random affine transforms, random Gibbs noise, and random Gaussian smoothing. The batch size used was 16, the learning rate was 0.001, and the optimizer was AdamW. These parameters were experimentally chosen. The validation AUC was fluctuating around 0.6, and the validation loss around 0.7. The final trained model behavior shows that employing SimCLR to pretrain the model did not lead to significant improvement. Appendix B presents how the SimCLR model works and the learning curves for the final downstream experiment.

### 4.3. Is Vision Transformer's attention mechanism capable of predicting the methylation status?

Recently, Dosovitskiy et al. (Dosovitskiy et al., 2020) proposed Vision Transformers (ViTs) to perform tasks such as image classification. ViT is a concept adapted from transformers in the NLP domain. ViTs uniqueness comes from the attention mechanism it employs, which is utilized to capture long-range relationships and provide insight into what the model is focused on. Hence, in this experiment we employed ViTs to predict the methylation status. OPTUNA framework (Akiba et al., 2019) was used to optimize the hyperparameters of the ViT model. A batch size of 16, a learning rate of 0.007 and Adam optimizer were found as the best combination of hyperparameters. A summary of the rest of the hyperparameters are listed in Appendix C.

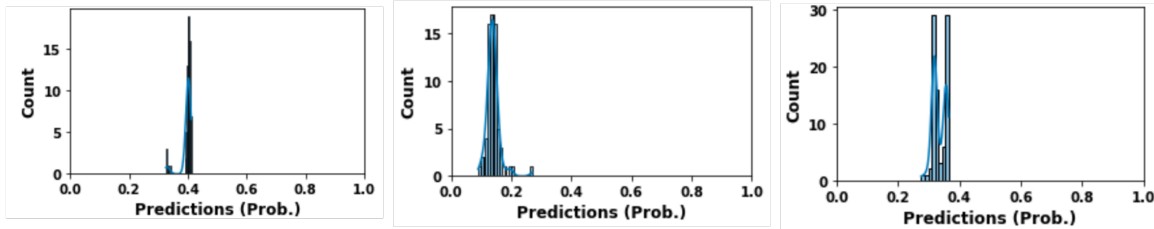

Figure 2: Histogram and Kernel Density Estimation (KDE) of predicted classification probabilities of three ViT models trained with different seeds.

As for the differences in the number of slices across patients, we chose the slice that shows the maximum cutaway of the brain as the central slice with 32 additional slices on each side. The MRI scans were resized to $256 \times 256 \times 64$. The validation AUC was found to be 0.58.

Since simple and complex models resulted in a validation accuracy no more than 0.6, the ViT model was trained with random seeds. Figure 2 shows the distribution of predictions for three of these models. As can be observed, the distribution of prediction on test set is far from being bi-modal which suggests discrimination between the two classes is very poor. The same experiment was conducted for an additional six ViT models with different seeds, a similar pattern of prediction is observed (Appendix C).

## 5. Discussion

The accuracy of all the models attempted during this study was very low. Hence, the focus was to understand the performance of these models and the models that won the RSNA-MICCAI competition. The following experiment, along with the remainder of the experiments in this manuscript, was based on ResNet10 model. The choice of this model is motivated by the winning solution of the Kaggle competition and due to its simplicity, which makes it ideal for exploring multiple experiments.

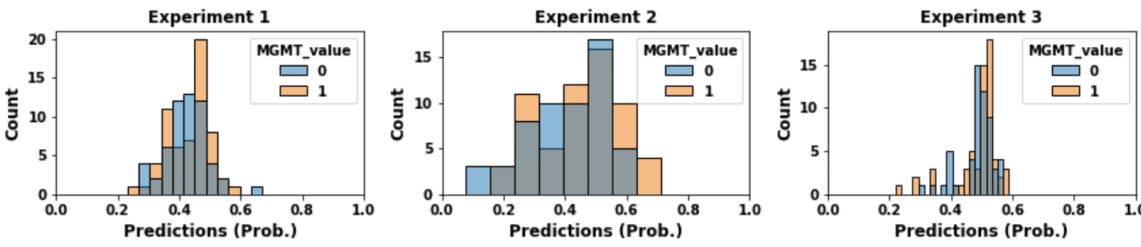

Figure 3: Histogram of the predicted classification probabilities conditioned on true labels.

It was found that the top three models in the competition, when used for inference on the test set, show a similar uni-modal distribution as discussed in the previous section. Unfortunately, the test labels for this dataset are not available publicly. Hence, part of the training data was reserved as a validation set while ensuring stratified sampling in the

Table 2: Class distribution of training set

| Class | Training | Validation |
|-------|----------|------------|
| 0 | 222 | 56 |
| 1 | 246 | 61 |

Table 3: The performance of 5 randomly selected ResNet10 models

| Model | 1 | 2 | 3 | 4 | 5 |
|-------|------|------|------|------|------|
| AUC | 0.57 | 0.59 | 0.54 | 0.58 | 0.57 |

training and the validation sets as per Table 2. The model was trained using ten different random seeds and then tested on the validation set. The validation accuracy of five of these models is listed in Table 3, and the rest of the validation AUC are in Appendix D.2. It is evident that the performance of all ten models is slightly better than a random guess. Figure 3 shows the prediction distribution of three of these models colored based on the true labels; the grey color shows the overlap between the two classes. The figure shows a significant overlap in the prediction distribution, suggesting that the model's prediction is primarily random. In addition, the mean of prediction varies significantly with the change of the seed. This behavior can be visually illustrated in Figure 5, which shows that all these models perform equally random because they cannot differentiate between the two classes. Additionally, this behavior indicates that the models are learning irrelevant noise and are incapable of finding predictive features.

To confirm the above hypothesis, the training loss curves for our models were observed. The training loss is initially high and saturates to an average value of 0.7 towards the end of the training, as shown in Figure 4. This trend was observed in our models and many other models on Kaggle. To explain this behavior, the binary cross entropy (BCE) loss Equation (1) is solved under the assumption that both classes are almost equally represented in the training set, and the model is in a random state, i.e., predicting both class 0 and 1 with probability 0.5. Solving Equation (1) yields a loss of 0.69, which explains that these models are in random states even after 15 epochs.

$$Loss = -\frac{1}{N}\sum_{i=1}^{N} y_i \times \log(p(y_i)) + (1 - y_i) \times \log(1 - p(y_i)) \tag{1}$$

These findings are contrary to some previous work; however, we cannot investigate the discrepancy in the performances due to the absence of public source code (Appendix D.1).

## 5.1. Interpretability

To interpret the trained ResNet10 model, the feature maps of a batch size of 128 were extracted at each layer, as shown in Appendix D.2. It was challenging to interpret the feature maps; therefore, we used t-SNE (Van der Maaten and Hinton, 2008) to plot them at every layer in a 2D space. It can be observed that towards the final layers, the two classes are still entangled, suggesting that the model is unable to find features that will differentiate between the two classes, as can be observed from Figure 6. As shown in Appendix D.2, the feature maps were extracted from the same model trained on MNIST-10. It is clear that the model can learn predictive features and can separate between the ten classes towards the final layers of the model. This experiment also confirms the hypothesis that there is no difference between the features extracted from the two classes' data.

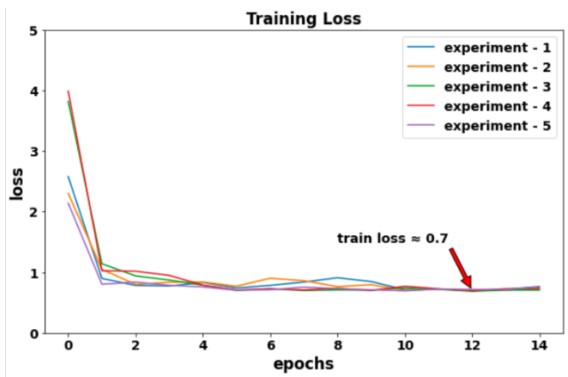

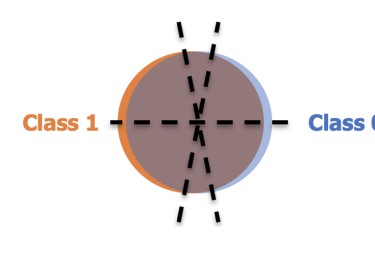

Figure 4: Training loss curves for different models.

Figure 5: Class overlap with random splits representing decision boundaries

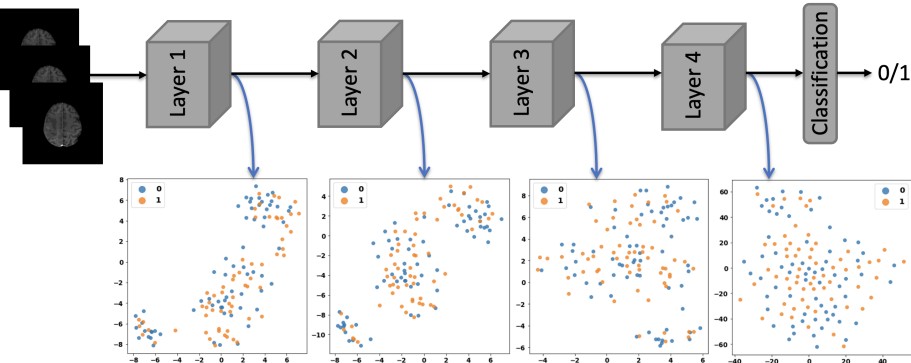

Figure 6: Feature visualization via t-SNE. The diffused points show that the model is not able to separate the two classes.

## 6. Conclusion

Finally, to answer the question posed in the title: although they showed promising performance in many medical applications, we believe that the current deep learning models and datasets cannot predict MGMT promoter state using only patient MRI scans. In this holistic study, we conducted a large number of experiments with different deep-learning architectures, but they all yielded an AUC that is not significantly better than a random guess for a binary classification problem. Models with this kind of performance cannot be used in clinical settings, and hence, as we stand, there is no alternative to surgical biopsies. However, these results can be altered in the future based on the development of novel datasets and methodologies. Also, it might still be possible to predict the methylation status by combining other biomarkers or prognostic factors. The future effort shall be focused on finding these biomarkers, creating new datasets, and developing novel methodologies.

## 7. Acknowledgements

The authors would like to thank Dr. Salman Khan and Dr. Karthik Nandakumar for their valuable feedback and insight, particularly on the interpretability section.

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

## Appendix A. Supplementary Details for CNN Methods

Table 4 shows the results for extended list of experiments.

Table 4: Results of the main experiments with 2D CNNs. Initialization: 0: Random, 1: ImageNet weights, Aug:Augmentations. * - as two channels.

| Model | Init. | Modality | Img Res. | Aug. | Train AUC | Val. AUC |
|---|---|---|---|---|---|---|
| ResNet34 | 0 | FLAIR | 128 | No | 0.95 | 0.51 |
| ResNet34 | 0 | FLAIR | 256 | Yes | 0.83 | 0.54 |
| ResNet34 | 1 | T1wCE | 128 | Yes | 0.51 | 0.55 |
| Resnet34 | 1 | T1wCE | 256 | No | 0.99 | 0.52 |
| ResNet50 | 1 | FLAIR | 128 | Yes | 0.61 | 0.58 |
| DenseNet121 | 1 | FLAIR | 128 | Yes | 0.63 | 0.58 |
| DenseNet161 | 1 | T1wCE | 128 | Yes | 0.51 | 0.50 |
| Custom ResNet | 1 | Stacking | 128 | Yes | 0.76 | 0.63 |
| Custom ResNet | 0 | Addition | 128 | Yes | 0.50 | 0.49 |
| ROI ResNet18 | 0 | FLAIR | 128 | No | 0.97 | 0.55 |
| ROI ResNet18 + more FC | 1 | T1wCE | 128 | Yes | 0.88 | 0.55 |
| ROI Custom ResNet | 0 | Combined* | 32 | No | 0.70 | 0.53 |

Architectural design of the model for ROI-based classification can be found in Figure 7.

Figure 8 depicts the architecture of the custom ensemble network for utilizing information from two MRI modalities.

## Appendix B. Supplementary Details for Self-Supervised Learning Methods

Figure 9 shows the SimCLR self-supervision procedure in the proxy task. In Figure 10, the learning curves of the downstream task of SimCLR are presented. Both stages use the ResNet-18 model.

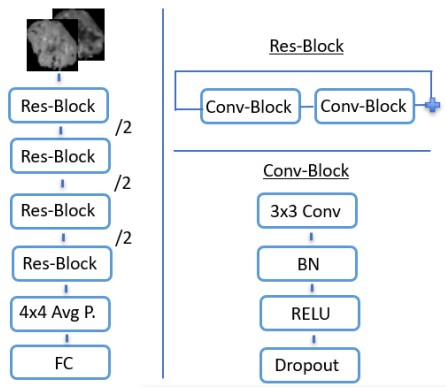

Figure 7: Architecture of the custom ResNet for ROI-based classification, as described in (Chang et al., 2018). Each residual block is comprised of two convolutional blocks with added dropout layer. After each stage, feature maps are downsampled by applying convolution with stride 2 (indicated by /2). $4 \times 4$ Average pooling is used before FC layer.

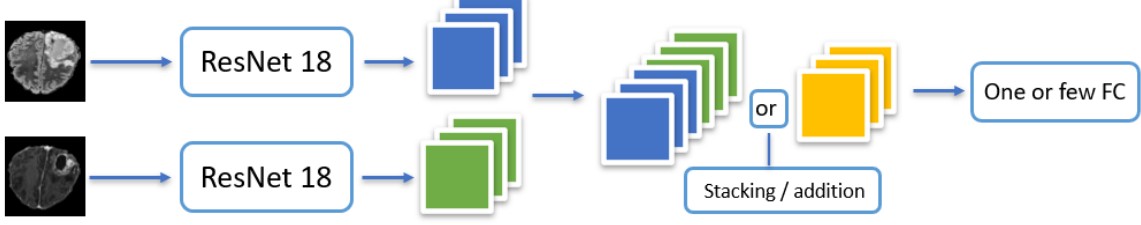

Figure 8: Architecture of the custom ResNet for combining FLAIR and T1wCE modalities.

## Appendix C. Supplementary Details for Vision Transformer Methods

Table 5 shows the hyperparameter values for the experiments with ViT. These values were found after running multiple experiments using OPTUNA framework (Akiba et al., 2019).

Table 5: ViT hyperparameter values

| Hyperparameter | Value |
|---|---|
| Batch size | 16 |
| Learning rate | 0.007 |
| Patch size | 32 |
| Embedding dimension | 2048 |
| Model depth | 2 |
| Multi-heads | 8 |
| Head dimension | 1024 |
| MLP dimension | 512 |

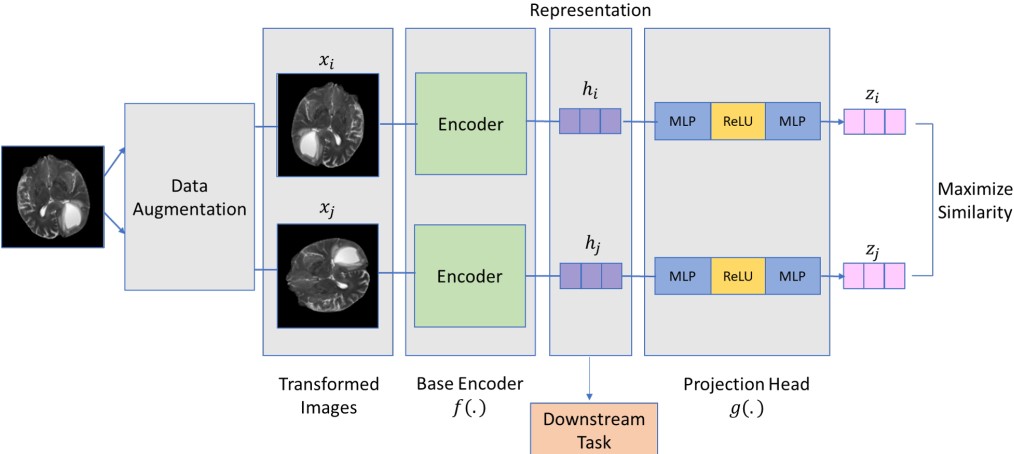

Figure 9: How SimCLR works: Two augmented versions of the same image are taken by the model and passed to the encoder (ResNet-18). Then, the representations of the encoder are passed to a non-linear projection head. The goal of SimCLR is to maximize the similarity of the vectors resulted from the projection head using a contrastive loss. The downstream task uses the representations from the encoder.

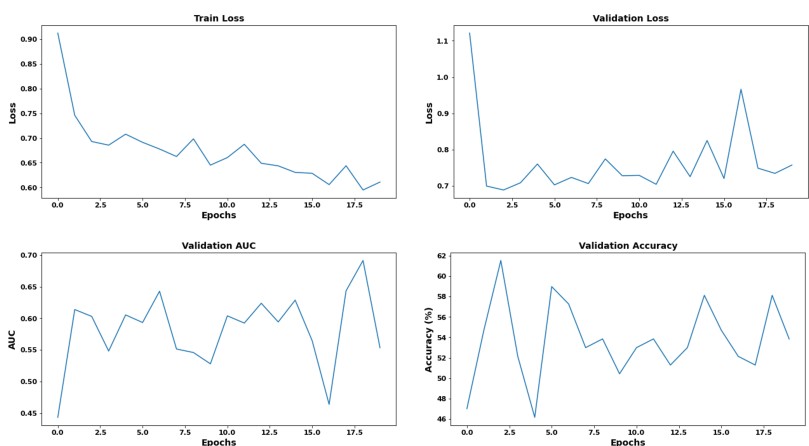

Figure 10: Learning curves for the SimCLR downstream task.

Figure 11 shows the prediction distribution for different ViT models.

## Appendix D. Supplementary Details for Discussion and Conclusion

### D.1. Speculation of Discrepancy in Results

We believe there are many possible reasons for the existing discrepancy in the results reported in the referenced papers. Some of the critical factors for higher performance can be the small dataset and the lack of a rigorous validation process. For example, (Chang

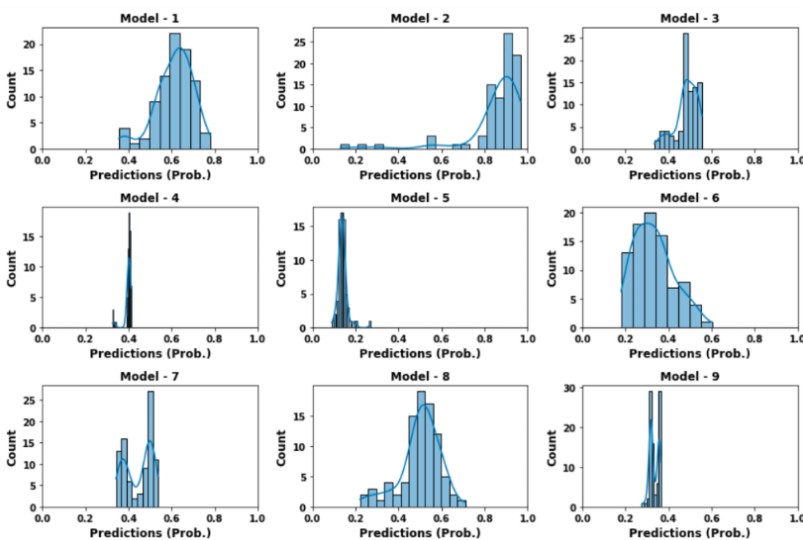

Figure 11: Histogram and Kernel Density Estimation (KDE) of predicted classification probabilities for ViT models trained with different seeds

et al., 2018), who got a high accuracy, outlined in their discussion that the relatively small sample size ($N = 259$) and the absence of an independent dataset might have affected the assessment of the generalization performance of their developed model. Similarly, another work by (Korfiatis et al., 2017) highlighted that their data were acquired from a single source, which should be avoided to produce unbiased results. (Yogananda et al., 2021), who also reported a high accuracy, described possible data leakage issues in their previous work. It is worth noting that their own work has a low segmentation accuracy but high MGMT-promoter status classification accuracy. Other researchers tried to reproduce the results in (Yogananda et al., 2021), but could not achieve the claimed performance and were unsuccessful in getting any response from the authors. We also tried to reimplement the strategies of the aforementioned authors by utilizing similar strategies of combining modalities using segmentation masks to perform region-of-interest-based classification. But, still, no improvements have been noticed. We cannot reproduce the results reported by these authors as their code is not publicly available.

The results of any study can be altered in the future based on the development of novel datasets and methodologies. Thus, it might still be possible to predict the methylation status by combining other biomarkers or prognostic factors. To generalize this situation, the question of using high-accuracy models in a clinical setting always remains one of the most pertinent ones. Our study reminds the AI-centered scientific community about the importance of thorough and unbiased validation for future studies and clinical implementation. We believe our contribution encourages extending the research ground to investigate other prognostic biomarkers and raise ethical concerns by giving a clear example.

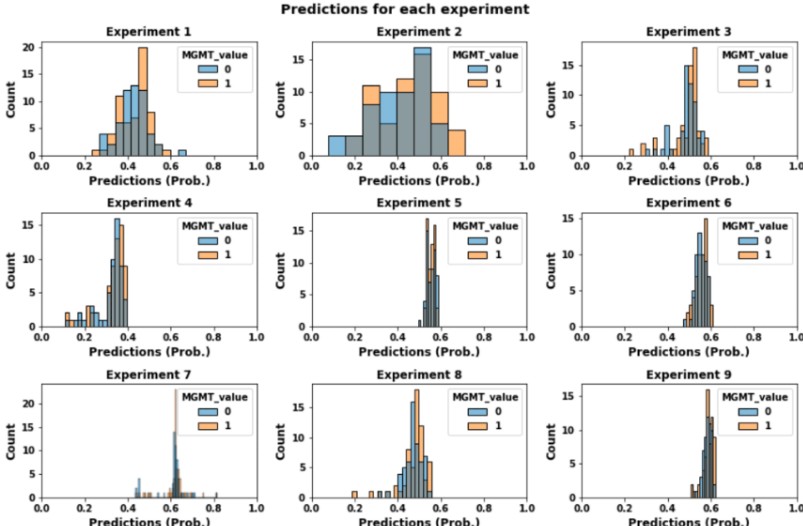

Figure 12: Histogram of predicted classification probabilities of ResNet10 models conditioned on true labels

## D.2. Further Model Analysis

Figure 12 shows the predicted probabilities distribution for nine ResNet10 models which were trained using different random seeds. Figure 13 shows the feature maps extracted at each layer of ResNet10, when inferred using an MRI scan of a patient. Figure 14 shows the feature maps of MNIST data extracted at each layer of ResNet10 and projected onto a two dimensional space using t-SNE. The model is able to perfectly separate the features for the different class compared to the MRI scans dataset.

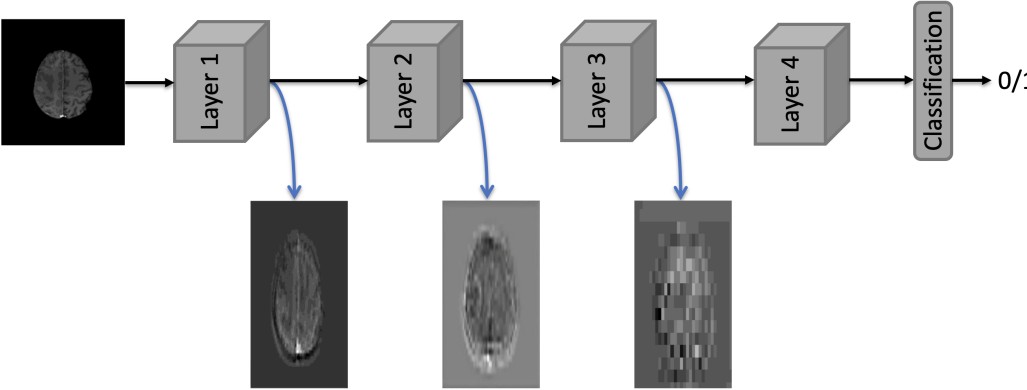

Figure 13: Feature visualization at different layers

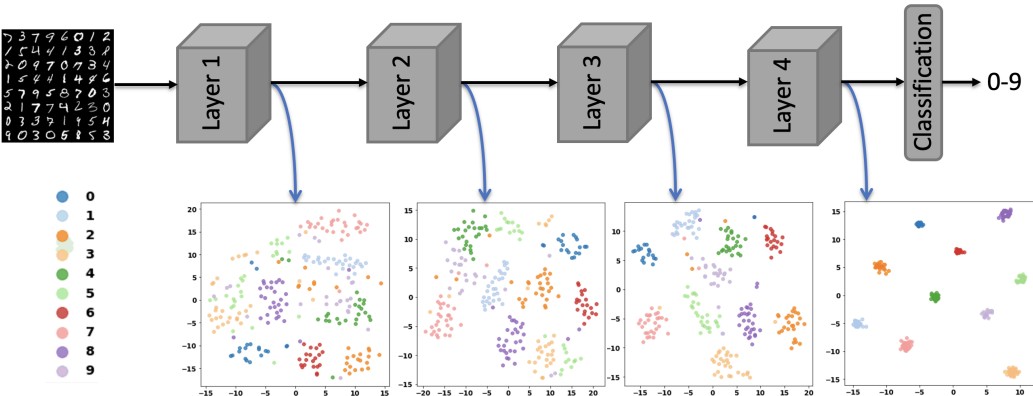

Figure 14: Feature visualization via t-SNE for MNIST data

