# OpenReview forum: "Is it Possible to Predict MGMT Promoter Methylation from Brain Tumor MRI Scans using Deep Learning Models?"
_MIDL.io/2022/Conference — MIDL 2022_

### Official Review · Reviewer_CFyt · 2022-01-22

**Confidence:** 4
**Preliminary Rating:** 3
**Recommendation:** Poster

**Summary:**

The work focuses on the validation of applying deep learning related models to predict the MGMT promoter state using MRI. The author considered 2D, 3D convolutional neural networks and a recent vision transformer model. From the analysis of the results of the experiments on the BraTS 2021 dataset, the authors conclude that it seems the deep learning models are not able to predict the state of MGMT promoter from MRI scans using deep learning.

**Strengths:**

* Well written and easy to follow, and it provides sufficient figures for illustrations.
* The related work on MGMT status prediction is comprehensive. However, I suggest the authors also discuss other related works about image-based gene expression prediction on various diseases.


**Weaknesses:**

* The technical contribution of this paper is limited. In recent years, there are quite a lot of papers that deal with the prediction of gene expression based on images using deep learning models, and this paper is among this category. They simply differ in the specific diseases. I think this work is more suitable for a clinic-oriented journal.

* As we know, if the cross entropy is around 0.7, it suggests the predicted binary probabilities for both categories are around 0.5, as the cross entropy is 0.69 if the probabilities are exact 0.5. Further, given the AUC values are all under 0.6, it seems the results are simply random guess. These two metrics are enough to conclude that the models the authors adopted failed to predict the status. Therefore, I don’t see it necessary to provide more redundant evaluations (e.g., t-SNE )


**Deanonymize Review:**

no

**Final Rating After The Rebuttal:**

4: Weak Accept

**Justification Of The Final Rating:**

Thank you for the reply. My concerns are partly addressed by the authors, but I see no major flaws with this work, and I agree with the clinical contribution. Although this failed attempt for this specific application with deep learning techniques may be restricted only to the cohort of dataset used, the experience can provide a good reference for the following works. Therefore, I upgrade to 'weak accept'.

**Paper Type:**

validation/application paper

**Questions To Address In The Rebuttal:**

* The self-supervision part in Pag.4-page.5,it is not clear why the authors pretrain the backbone model with another dataset. Can’t just learn a feature extractor with the same dataset using SimCLR ? Could the author provide the motivation, please?

**Special Issue:**

no

---

### Official Review · Reviewer_TsCy · 2022-01-24

**Confidence:** 3
**Preliminary Rating:** 4
**Recommendation:** Oral, Poster

**Summary:**

The prediction of MGMT promoter methylation status based on MRI scans of glioblastoma has been investigated in recent years as a non-invasive proxy to predict response to chemo therapy, with diverging results. In this paper, the authors use the BraTS 2021 dataset to extensively study different deep learning architectures and training strategies for this task, including 2-D and 3-D CNN models, vision transformers and self-supervised learning. For none of the trained models, a relevant predictive performance can be established, with the highest performing models reaching an AUC score of approx. 0.6. Based on their results, the authors conclude that MGMT promoter methylation status cannot be derived from MRI scans.


**Strengths:**

The authors present an extensive and systematic analysis of different network architectures and learning strategies for the aforementioned task, and clearly report that they do not find any evidence of association between methylation status and MRI image features (negative findings are important and often underrated!). The paper is generally well written and well structured. In the related work section, diverging reports in the literature are well summarizes, with papers reporting a connection and other papers not able to find an association. The discussion on the discriminative power of a comparatively simple network adds to the point that the networks are not able to derive discriminatory features on the methylation status from MRI.

**Weaknesses:**

The paper reads like an extension of the BraTS 2021 challenge, with a more systematic evaluation of models used for the task. Although I see the value in this systematic approach, it adds little to the general evidence of the challenge results. It feels like a lost opportunity not to look deeper into why diverging results have been reported in literature to identify potential issues in study design, network training etc. This could potentially further underscore the validity of the presented results and help researchers to identify similar issues (if present) in future studies.

**Deanonymize Review:**

yes

**Detailed Comments:**

- I am not quite sure how to put this or whether this is rather a personal feeling - but the paper seems to take some aspects from a summary paper of the BraTS 2021 challenge - if not done already, it would be good practice to shortly announce this and potentially discuss this with the organizers of the challenge.
- Misleading: Egana et al. don't look at MRI scans, but only look at the relationship between methylation status and treatment response.
- Preprocessing is mentioned both in Datasets and in Experiments - this should be homogenized
- Sec. 4.2: The description of the data extraction for the self-supervised task and the downstream task could potentially be homogenized and clarified, e.g., it is not clear why for the network pretrained in a self-supervised fashion, only the FLAIR modality was used, what do the authors mean by "used them as a 20 channel 2D input to the model", and why the central slice and not the tumor region used as input?
- It is interesting to add the point of a random prediction to the discussion (pg. 7), but the training AUC is for a subset of experiments on the larger models between 0.8-0.9 (indicating overfitting). Here, a clarification would be good - was the training loss considerably lower for these cases?

Minor suggestions + typos:
- Update Yogananda et al. reference to AJNR publication (pg. 2)
- Chang et al. 2018 is mentioned twice (pg. 2)
- Missing capitalization of abbreviations in the literature list should be corrected
- et. al. > et al. (pg. 2)
- why is accuracy (and most other measures) reported in percent, but precision and recall as decimal numbers? (pg. 2)
- T1wCE is not defined - do the authors mean T1Gd (pg. 3)
- "variations of 3D architecture [...] were used to train the models" > expression could be improved (pg. 4)
- The description section 4.2 could be read such that BraTS 2020 does not have an overlap with the BraTS2021 dataset - is this the case?
- Batch size seems to be rather large with 128 images / batch - this could (partly) explain the overfitting. Did the authors also experiment with smaller batch sizes?
- "As shown in Appendix C, the feature maps were extracted from the same model trained on MNIST-10." > a bit unclear, which point do the authors want to make here? (pg. 8)
- "As in this holistic study" > "In this holistic study" (?)
- There seem to be some inconsistencies in the plots (e.g., different width of histogram bars, potentially deformed feature maps (not square)), also: it could be shortly mentioned what the gray parts of the bars mean (Fig. 12).

**Final Rating After The Rebuttal:**

4: Weak Accept

**Justification Of The Final Rating:**

The authors answered the questions raised in the rebuttal satisfactory and the points raised in this paper may also foster an interesting discussion on tackling publication bias, for which an oral presentation could present an appropriate platform.

**Paper Type:**

validation/application paper

**Questions To Address In The Rebuttal:**

Findings can always be revised or nullified based on additional studies, a larger cohort, a clean/corrupted dataset. Still, given that the prediction performance for MGMT promoter methylation was very high in a set of prior studies, an investigation or discussion of this mismatch and discordance would be of high interest to the scientific community (e.g., confounders in TCIA). At the moment the paper makes a bold (though likely valid) statement but leaves these points open, therefore only adding evidence to one side of the discussion. Apart from being systematic (which is appreciated), it does not add much to the results of the BraTS challenge. A discussion of this point by the authors in the rebuttal would be appreciated.

Minor point, given the overfitting / underfitting issues: For many experiments, a rather large batch size was used (also w.r.t. the size of the training data set). Did the authors observe a different behaviour for smaller batch sizes?

**Special Issue:**

no

---

### Meta-Review · Area_Chair_mLYH · 2022-02-16

**Recommendation:** Accept (Poster)
**Confidence:** 5

**Metareview:**

This paper questions the feasibility of predicting MGMT status from MR image alone, despite many positive results reported by others. All reviewers recognized the importance of reporting negative results and the quality of the paper. Although methodological contributions are limited, I recommend the acceptance of the paper.

As pointed out explicitly (rev. UBUu) or implicitly (rev. TsCy), I would recommend incorporating a discussion on why other authors did in fact report a higher performance for a similar task (even in an appendix) for the camera ready version. The reasons mentioned in the rebuttal should be used.

---

### Decision · Program_Chairs · 2022-02-28

Accept